# BrainODE: Dynamic Brain Signal Analysis via Graph-Aided Neural Ordinary Differential Equations

Kaiqiao Han
*Dept. of Computer Science*
*University of California, LA*
Los Angeles, USA
kqhan@cs.ucla.edu

Yi Yang
*Dept. of Computer Science*
*Duke University*
Durham, USA
owen.yang@duke.edu

Zijie Huang
*Dept. of Computer Science*
*University of California, LA*
Los Angeles, USA
zijiehuang@cs.ucla.edu

Xuan Kan
*Dept. of Computer Science*
*Emory University*
Atlanta, USA
xuan.kan@emory.edu

Ying Guo
*Dept. of Biostat. & Bioinf.*
*Emory University*
Atlanta, USA
yguo2@emory.edu

Yang Yang
*Dept. of Computer Science*
*Zhejiang University*
Hangzhou, China
yangya@zju.edu.cn

Lifang He
*Dept. of Computer Sci. & Eng.*
*Lehigh University*
Bethlehem, USA
lih319@lehigh.edu

Liang Zhan
*Depts. of ECE and BioE*
*University of Pittsburgh*
Pittsburgh, USA
liang.zhan@pitt.edu

Yizhou Sun
*Dept. of Computer Science*
*University of California, LA*
Los Angeles, USA
yzsun@cs.ucla.edu

Wei Wang
*Dept. of Computer Science*
*University of California, LA*
Los Angeles, USA
weiwang@cs.ucla.edu

Carl Yang
*Dept. of Computer Science*
*Emory University*
Atlanta, USA
j.carlyang@emory.edu

*Abstract*—**Brain network analysis is vital for understanding the neural interactions regarding brain structures and functions, and identifying potential biomarkers for clinical phenotypes. However, widely used brain signals such as Blood Oxygen Level Dependent (BOLD) time series generated from functional Magnetic Resonance Imaging (fMRI) often manifest three challenges: (1) missing values, (2) irregular samples, and (3) sampling misalignment, due to instrumental limitations, impacting downstream brain network analysis and clinical outcome predictions. In this work, we propose a novel model called BrainODE to achieve continuous modeling of dynamic brain signals using Ordinary Differential Equations (ODE). By learning latent initial values and neural ODE functions from irregular time series, BrainODE effectively reconstructs brain signals at any time point, mitigating the aforementioned three data challenges of brain signals altogether. Comprehensive experimental results on real-world neuroimaging datasets demonstrate the superior performance of BrainODE and its capability to address the three data challenges.**

*Index Terms*—**Neural ODEs, Brain signal analysis, Graph neural networks**

This research was supported by the National Science Foundation under Award Number 2319449 and Award Number 2312502. Lifang He is partially supported by the NSF grants (IIS-2319451, MRI-2215789), NIH grant R21EY034179, and Lehigh's grants under Accelerator and CORE. Liang Zhan is partially supported by the NSF grants (IIS-2319450, IIS-2045848) and NIH grants (RF1 MH125928, U01 AG068057). Carl Yang is partially supported by the NIH grant K25DK135913.

## I. INTRODUCTION

Recent advancements in neuroimaging techniques, such as the prominent development of functional Magnetic Resonance Imaging (fMRI), have greatly propelled neuroscience research and brain connectome analysis. Specifically, the utilization of fMRI scans has enabled the creation of functional brain networks, where nodes are composed of anatomical regions of interest (ROIs), and links are derived from the correlations among the Blood Oxygen Level Dependent (BOLD) time series associated with the ROIs. By effectively modeling such functional correlations among the BOLD signals, researchers gain deeper understandings of the functions and organizations of complex neural systems within the human brain, which can help derive valuable clinical insights for diagnosis of neurological disorders [1]–[3].

Brain network analysis typically relies on a tedious pipeline for brain imaging preprocessing and ROI signal extraction [4]. For functional brain networks, the pipeline is mainly focused on the processing of fMRI into BOLD signals associated with ROIs, while the subsequent construction of functional brain networks often simply uses the direct computation of Pearson correlations based on the raw BOLD signals, largely overlooking the dynamic nature of BOLD signals as well as their limited data quality [5], [6]. This can further lead to inaccurate network modeling and misleading downstream predictions, carrying significant ramifications for the comprehension of

brain networks and their potential clinical applications [7], [8].

In this work, we model brain signals as dynamic time series and identify three commonly encountered data challenges that should warrant particular attention (as illustrated on the left side of Figure 1). (1) Missing values. The neuroimaging collection might be missing at certain time points due to abrupt fluctuations and mechanical noises. (2) Irregular samples. The machine may not collect the data precisely at the desired time points. For example, the machine is expected to sample at time point 1s, but the actual sample is collected at time point 1.2s due to instrumental errors. (3) Sampling misalignment. Different samples may be collected under different frequencies. For example, a machine collects the data every second while another collects it every two seconds.

There exist simple interpolation methods such as those based on mean values or polynomial regression to partially address the data challenges [9]. However, they are not ideal for capturing the complex dynamics of ROIs in brain networks [10]. In recent years, Neural Ordinary Differential Equations (ODEs) have emerged as a powerful framework for irregularly-sampled data and dynamic systems, which has proven immense successes in applications to physical system simulation and disease spread modeling [11], [12]. However, existing Neural ODE methods do not consider implicit interactions among brain signals, which is essential in brain imaging analysis [4].

Inspired by neural ODE, we propose to conquer the three data challenges for brain signal analysis collectively with a unified re-processing procedure for dynamic brain signals, by learning a continuous model of interconnected signals over time, which can regulate and reconstruct the signals at any particular time point. Specifically, we develop a novel model called BRAINODE to learn the latent initial states and neural ODE functions automatically from the partially available dynamic brain signals, which can continuously reconstruct the brain signals at any given time point. In addition, to estimate the most informative latent initial states of ROIs, we propose to construct two graphs to capture the two most important types of ROI relations in brain networks– structural (spatial) and functional (temporal). We apply graph convolutional networks (GCNs) on the two graphs and leverage the combined representations to enhance the learning of initial states for effective ODE inference. These two graphs not only utilize common wisdom about brain connectivities in neuroscience research, but also provide potential opportunities for deriving clinically actionable discoveries. By mitigating the impact of low data quality, BRAINODE enhances the usability of dynamic brain signals in clinical predictions with improved performance for various downstream brain network analysis models. In summary, our key contributions are as follows.

- We are the first to recognize the importance of re-processing dynamic signals in brain network analysis and identify three major data challenges. We provide a unified framework to address the challenges through continuously modeling the interactive brain signals.

- Inspired by the recent success of neural ODE, we propose BRAINODE as the unified solution. We utilize spatial and temporal graphs to capture the complex interactions among ROIs, thereby facilitating continuous Neural ODE learning and inference.
- We conduct experiments on two real-world datasets to verify the effectiveness of the proposed model. The AUC performance of classification increases *avg.* 15.6% in ABIDE and *avg.* 27.4% in ABCD. Additionally, we conduct further experiments to demonstrate the model's efficacy in individually solving the three aforementioned data challenges with superior performance.

Our work can be used for brain analysis and potentially be employed to search for associations between brain signals and clinical outcomes such as mental disorders. To safeguard against any misuse or unauthorized exploitation, we advocate for the responsible and cautious use of our findings. We emphasize that the utilization of our work should be limited exclusively to ethical and peer-reviewed academic research.

## II. BACKGROUNDS AND RELATED WORKS

*1) GNNs for Brain Network Analysis:* Graph Neural Networks (GNNs) have garnered significant interest for their effectiveness in analyzing graph-structured data [13]–[17], which stimulates studies of their application in brain connectome analysis [4], [8], [18]–[21]. While showing great promises of GNNs in enhancing the performance of brain connectome based clinical outcome predictions, the current data-driven studies for neuroimaging and brain network analysis are mostly based on pre-constructed brain networks, which lacks adequate focus on data pre-processing and its influence on downstream performance.

*2) ODE for Multi-Agent Dynamical Systems:* A multi-agent dynamical system can be captured and reconstructed continuously by a series of first-order Ordinary Differential Equations (ODEs), which describes the continuous evolution of $N$ dependent variables in the time window $[0, T]$ [11]. For each object $i$ in time $t$, its state can be uniquely determined by $z_i^t$ and each object has the corresponding ODE: $\dot{z}_i^t := \frac{dz_i^t}{dt} = g(z_1^t, z_2^t, \ldots, z_N^t)$. Given the latent initial states for every object, we can use a numerical ODE solver such as Runge-Kutta to get the state at any given time point [22].

Recently, abundant studies have advocated the learning of ODE functions $g_i$ using neural networks [23], [24]. As closest to us, [12], [25] propose GraphODE to jointly model the evolution of entities and their connections by combining GNNs and neural ODE. However, GraphODE needs the given integral dynamic graph structure to utilize the relation of agents, so it cannot be directly applied to brain signal analysis due to the lack of ground-truth dynamic graph structure to facilitate model learning. Also, the existing ODE methods are unable to handle multiple types of entity interactions, such as brain function network and brain structure work. Capturing the complex relationships among ROIs and integrating them with neural ODE remain a challenging task.

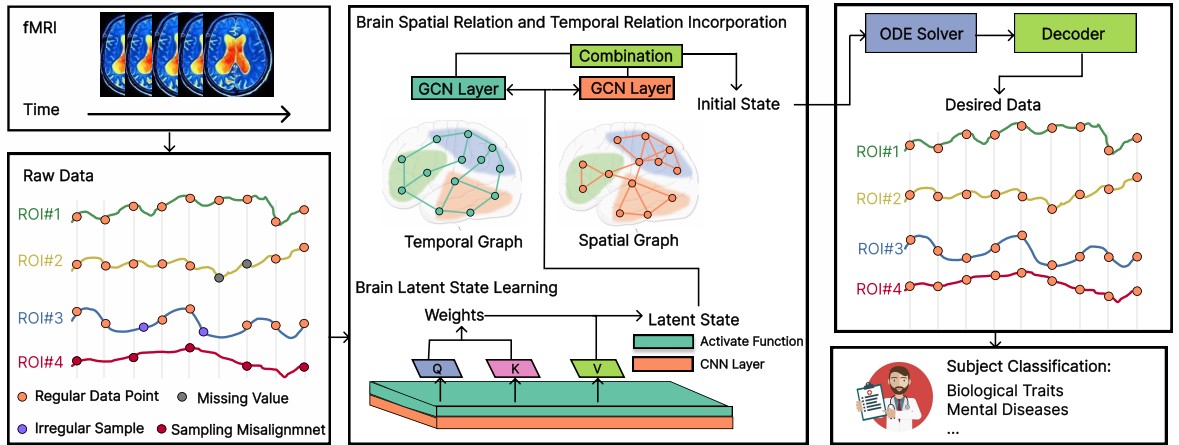

Fig. 1: Overview of BRAINODE. The left panel demonstrates the raw dynamic brain signal from the fMRI sequence where ROI#1 shows an ideal signal and ROI#2-4 illustrate the "missing value", "irregular sample", and "sampling misalignment" challenges, respectively. The middle panel describes the BRAINODE framework which leverages Short-Term and Long-Term Time Encoders to learn a latent embedding for each input ROI signal. The embeddings are then refined through two distinct GCN layers, which learn the spatial and temporal relations among ROI channels, aiding in the derivation of initial states for ODE inference. An ODE solver and decoder, shown on the right panel, are utilized to obtain the brain signals at desired regular time steps, which can lead to enhanced performance in subject classification.

## III. METHOD

### A. Problem Definition

We focus on the problem of continuously modeling the dynamic brain signals (*e.g.,* BOLD time series). We consider a dataset consisting of $S$ samples (subjects) with each representing a distinct brain network neuroimage. Each subject can be parcellated into $N$ ROIs, each corresponding to a time series of signals with a length of $T$. Therefore, a dataset can be represented by a tensor of signals $X \in \mathbb{R}^{S \times N \times T}$, and its associated time coordinates $T_s \in \mathbb{R}^{S \times N \times T}$. BRAINODE reprocesses the input data $X$ to address the problems of missing values, irregular samples, and sampling misalignment. The outcome is a desired set of signals $X_R \in \mathbb{R}^{S \times N \times T'}$ along with their respective time coordinates $T_R \in \mathbb{R}^{S \times N \times T'}$, where $T'$ denotes the desired length of the signals. $X_R$ can be used in the same way as $X$ for brain network analysis and downstream predictions, potentially with improved accuracy due to the enhanced quality of dynamic brain signals. In this work, we follow the standard practice for brain network construction by computing Pearson correlations between the dynamic brain signals [6], and train existing brain network classification models towards given subject labels $Y \in \mathbb{R}^{N \times |C|}$, with $C$ denoting the class set and $|C|$ being the number of classes.

### B. Methodology Overview

We propose BRAINODE, a novel approach for continuous modeling of dynamic brain signals. The framework comprises three key components: Brain Latent State Learning, Brain Temporal Relation and Spatial Relation Incorporation, and Autoencoder-based End-to-End Training. The first two modules are designed for inferring the initial states of neural ODE and serve as the starting point for predicting the trajectories in the latent space. In particular, the latent state learning leverages a Convolutional Neural Network (CNN) to capture

the brain activity, and a self-attention mechanism to model the long-range temporal dependencies of brain signals. We then construct temporal and spatial graphs to represent the multi-facet relations among ROIs and use GCNs to encode the signals, under the aid of graph structures, for enhanced latent initial state learning. Finally, a generative model based on ODEs generates continuous representations of the time series data inferred from the learned initial states, which is also end-to-end trainable through an autoencoder framework.

### C. Brain Latent State Learning

Dynamic brain signals in the form of time series data convey rich information regarding the functional relations among ROIs [26], [27]. Recent research leverages the Pearson correlation measures to identify groups of ROIs that exhibit similar dynamics to help discover underlying neural patterns [28], [29]. However, similarity measures based on global time series tend to overlook the neural activation patterns in shorter time frames, which undermines the expressiveness in representing the dynamical changes in brains. Therefore, to encode a set of dynamic-aware temporal features, we propose a CNN-based encoder to capture brain activity, and a self-attention module to model the long-term (global) dependencies.

*1) Brain Activity Modeling:* CNNs have demonstrated remarkable capabilities in capturing intricate features from grid-like data and it is also widely used in analyzing sequential data via dilated convolution [30]. For each ROI signal, the feature map of a convolutional layer can be formulated as follows:

$$f_i^j = \sigma \left( \sum_{k=1}^{F} (Conv_k * RF_j(x_i)) + \beta \right), \quad (1)$$

where $x_i$ represents the initial signals of the $i$-th ROI, $Conv_k$ represents the convolution filter at the $k$-th layer, $RF_j(\cdot)$ is the Receptive Field of $j$-th time step where $j \in [1, T]$, $\sigma$ is the

activation function that introduces non-linearity to the output, and $\beta$ represents an additive bias.

*2) Brain Signal Long-term Dependency Modeling:* To prepare the initial values of the observed dynamic signals on ROIs for subsequent ODE inference, we devise a self-attention mechanism to capture the long-term dependencies among dynamic signals after the short-term temporal representations are learned. In particular, self-attention demonstrates impressive efficacy in simultaneously capturing sequential relations and mitigating catastrophic forgetting of long-range information [31]. For dynamic signals, the attention mechanism can be adapted to capture dependencies and relationships between entries across a wide interval of time steps.

After the convolution layer, we have the temporal representations for the $i$-th ROI, $f_i \in \mathbb{R}^{T \times F}$, that contains a set of $F$ dimensional encoding for every time step in a total of $T$ steps. We also leverage positional encoders [32], [33] to differentiate time steps and capture their sequential information. $\text{PE}_{(pos,2l)} = \sin\left(\frac{pos}{10000^{2l/d_{\text{model}}}}\right)$, $\text{PE}_{(pos,2l+1)} = \cos\left(\frac{pos}{10000^{2l/d_{\text{model}}}}\right)$, where $pos$ is the position of the time step, $l$ is the dimension of the positional encoding, $d_{\text{model}}$ is the dimensionality of the embedding size.

For applying self-attention, the input representation $f_i$ is transformed into three matrices: Query $Q$, Key $K$, and Value $V$, each of which is a linear projection of the original data: $Q = f_i W_Q, K = f_i W_K, V = f_i W_V$, where $W_Q$, $W_K$, and $W_V$ are learnable weight matrices used to project the input time series into query, key, and value spaces, respectively. The self-attention mechanism computes a similarity score between each query-key pair. This is typically achieved by using the dot product. At last, the initial value of every ROI is computed by the attention scores and the $V$ matrix:

$$h_i = \text{Softmax}\left(\frac{QK^\top}{\sqrt{d_k}}\right)V, \tag{2}$$

where $d_k$ is the dimension of the key space. The division by $\sqrt{d_k}$ is used to stabilize the gradients during training.

### D. Brain Temporal Relation and Spatial Relation Incorporation

Neuroscience research suggests that there are close connections between different regions of the brain manifested through both functional correlations and structural proximities [34]. Along this line, previous studies of brain network analysis have confirmed the necessity of simultaneously capturing the instantaneous excitation and inhibition of temporarily related ROIs as well as the anatomic structures of spatially neighboring ROIs [4], [20]. Therefore, we propose to explicitly model the temporal and spatial relations among ROIs by constructing two graphs that share the same set of nodes as ROIs (or equivalently one graph with two types of links), and use them to enhance the learning of our neural ODE model through additional encodings of the learned initial values.

*1) Temporal Graph Construction:* We aim to construct a temporal graph that captures the correlations among dynamic brain signals to aid the learning of BRAINODE. We learn the graph ODE functions based on it, which captures the variability across subjects and makes the framework more generalizable. However, the original dynamic brain signals can be irregular To this end, we propose to define the temporal structure based on the initial values of the ROIs. We note that, due to the joint, end-to-end learning of the temporal graph and the ODE function (the next component), our temporal graph encodes functional connectivity as defined in neuroscience literature, but it is fundamentally different from existing functional brain networks which rely on high-quality brain signals at first.

Specifically, given $N$ ROIs with their respective initial values $h = \{h_1, h_2, \ldots, h_N\}$, we calculate the cosine similarity between every pair and construct an adjacency matrix $A^{Tem}$ for the temporal graph as $A_{ij}^{Tem} = \frac{h_i \cdot h_j}{\|h_i\| \cdot \|h_j\|}$, where $A_{ij}^{Tem}$ represents the cosine similarity between the initial values of the $i$-th and the $j$-th ROI, and $\|\cdot\|$ denotes the $\ell_2$ norm.

*2) Spatial Graph Construction:* We leverage the 3D coordinates of ROI centers given by their corresponding parcellation templates to construct the spatial graph that encodes the structural connectivity of ROIs.

Given a set of $N$ ROIs and their 3D coordinates set: $\{(x_{1,1}, x_{1,2}, x_{1,3}), (x_{2,1}, x_{2,2}, x_{2,3}), \ldots, (x_{N,1}, x_{N,2}, x_{N,3})\}$, for each pair of ROI coordinate $(x_i, x_j)$, we calculate the Euclidean distance between them as follows: $Distance_{ij} = \sqrt{\sum_{d=1}^{3}(x_{i,d} - x_{j,d})^2}$. Next, we construct an adjacency matrix $A^S$ based on the Euclidean distances. The adjacency matrix indicates the existence of edges between ROIs based on a distance threshold $r$. $r$ is obtained through grid search. If the Euclidean distance between two ROIs is below the threshold, we consider them to be connected, and the corresponding entry in the adjacency matrix is set to 1. Otherwise, the entry is set to 0. Leveraging the fixed distance threshold $r$ as a sparsification mechanism, $A^S$ is computed as: $A_{ij}^{Spa} = \begin{cases} 1, & Distance_{ij} \leq r, \\ 0, & Distance_{ij} > r. \end{cases}$

*3) Relation-Aided Dynamic Brain Signal Modeling:* We adopt the graph message passing design proposed in GCN [17] to perform information aggregation based on the neighborhood structures defined by the temporal graph $A^{Tem}$ and the spatial graph $A^{Spa}$. Given the set of latent initial values $H$ as the ROI-wise feature set, the GCN convolution at layer $l$ can be formulated as

$$H^{(l+1)} = \sigma\left(\hat{D}^{-\frac{1}{2}} \hat{A} \hat{D}^{-\frac{1}{2}} H^{(l)} W^{(l)}\right), \tag{3}$$

where $H^{(l)}$ represents the node embeddings at layer $l$ (initially, $H^{(0)} = H$), $W^{(l)}$ is the learnable weight for layer $l$, $\hat{A} = A + I_N$ is the adjacency matrix. $A$ is to be instantiated as $A^{Tem}$ for temporal graphs and $A^{Spa}$ for spatial graphs, and a GCN model is learned on each graph. $\hat{D}$ is the degree matrix of the adjacency, where $\hat{D}_{ii} = \sum_j \hat{A}_{ij}$ (summing over the $i$th row of $\hat{A}$). $\sigma$ is the ReLU activation function.

Lastly, we use a linear transformation to combine the two sets of representations into $u_i$ for every ROI.

### E. Autoencoder-based End-to-End Training

With the representation of each ROI, we estimate the approximated posterior distribution where $Tr$ is a neural network translating the representation into mean and variance of $z_i^0$ and $\mu_{z_i^0}, \sigma_{z_i^0} = Tr(u_i)$:

$$q_\phi(z_i^0|x_1, x_2, \ldots, x_n) = N(\mu_{z_i^0}, \sigma_{z_i^0}). \quad (4)$$

A generative model defined by an ODE is used to get the latent state at every time step with latent initial state $z_i^0$ sampled from the approximated posterior distribution $q_\phi(z_i^0|x_1, x_2, \ldots, x_n)$ from the encoder:

$$z_i^0 \sim p(z_i^0) \approx q_\phi(z_i^0|x_1, x_2, ..., x_n). \quad (5)$$

A neural network is employed as the ODE function $g_i$ to model the continuous change of signals on each ROI:

$$z_i^0, ..., z_i^T = ODESolve\left(g_i, [z_1^0, ..., z_n^0], (t_0, ..., t_T)\right). \quad (6)$$

A decoder is then utilized to reconstruct the dynamic signals from the decoding probability $p(o_i^t|z_i^t)$ according to the formula $o_i^t \sim p(o_i^t|z_i^t)$.

We connect the encoder, generative model, and decoder in an autoencoder-based framework and jointly train them in an end-to-end fashion w.r.t. the MSE loss. A KL divergence is added to normalize the initial states. The MSE loss ensures that the model accurately reproduces the input time series, while the KL divergence term regularizes the initial states and stabilizes the training process [11]. The overall training process is performed end-to-end, allowing all components to be optimized together. This joint training ensures that the entire model is well-integrated and capable of efficiently capturing the underlying continuity and complex relationships of dynamic brain signals.

### F. Time Complexity Analysis

For BRAINODE, the time complexities of the short-term time encoder and long-term time encoder are $O(TF)$ and $O(T^2F)$, respectively, where $T$ is the length of time series and $F$ is the filter number of convolution. In spatial relation and temporal relation incorporation, the time complexities of graph construction and GNN are $O(N^2)$ and $O(NF)$, respectively, due to the sparse relations of ROIs [4]. The time complexity of the ODEsolver and decoder is $O(T')$ where $T'$ is the desired time length.

## IV. EXPERIMENTS AND RESULTS

In this section, we evaluate the effectiveness of the proposed model BRAINODE through extensive experiments. We aim to answer the following important research questions:

**RQ1:** How does BRAINODE contribute to performance improvements across various base models compared to competing approaches?

**RQ2:** How does BRAINODE perform in addressing missing values, irregular samples, and sampling misalignment?

**RQ3:** What is the fundamental functionality and impact of the individual components of BRAINODE?

**RQ4:** What is the impact hyperparameters?

**RQ5:** How does the efficiency of BRAINODE compare to those of its opponents?

### A. 4.1 Experiment Settings

*1) Datasets and Prediction Tasks:* We conduct main experiments on two real-world fMRI datasets. (a) Autism Brain Imaging Data Exchange (ABIDE): This dataset collects resting-state functional magnetic resonance imaging (rs-fMRI) data from 17 international sites, and all data are anonymous [35]. The dataset contains 1,009 subjects, with 516 (51.14%) being Autism spectrum disorder (ASD) patients. The ROI definition is based on Craddock 200 atlas [36]. ABIDE supplies generated brain networks that can be downloaded directly without permission request. However, multi-site data are collected from different scanners and non-neural inter-site variability may mask inter-group differences. We follow a train-test data split strategy proposed in [20] to alleviate this issue. (b) Adolescent Brain Cognitive Development Study (ABCD): ABCD supplies the largest publicly available fMRI dataset with restricted access [37]. The data we use in the experiments are fully anonymized brain networks with only biological gender labels. The dataset includes 7,901 subjects with 3,961 (50.1%) among them being female. The ROI definition is based on the HCP atlas [38]. To analyze the capability of BRAINODE in addressing irregular samples and sampling misalignment, We use randomly trigonometric functions to simulate brain signals at arbitrary times due to the lack of real continuous brain signals [39].

*2) Opponent Models and Base Models:* We compare BRAINODE with three opponent models: (a) Polynomial interpolation (denoted as Poly) is a method widely used in time series preprocessing. It fits a polynomial function with partial time series to obtain the values at arbitrary times. (b) Recurrent Neural Network (denoted as RNN) is a neural network designed for processing sequential data by maintaining a hidden state that retains information from inputs [40]. (c) TSTPlus (denoted as TSTP) is the state-of-the-art (SOTA) method for multi-variable time series processing based on the Transformer architecture and self-supervised learning [41]. In RQ2.2, since RNN and TSTP cannot recover time series with arbitrary offsets and frequencies, different interpolation methods are used, which are exponential function fitting (denoted as Exp) and logarithmic function fitting (denoted as Log).

To validate that BRAINODE can enhance the performance of different models after re-processing the brain signals, we run four representative base models on the functional brain networks constructed from the original and re-processed brain signals. They respectively represent the SOTA models in fixed network, learnable network, graph transformer, and orthonormal clustering (a) BrainNetCNN is a neural network model for connectome-based subject classification [18]. (b) FBNetGNN is the SOTA model for end-to-end functional brain network generation and subject classification based on dynamic brain

TABLE I: The performance of different base models (first column) across different dynamic data processing methods (second colum). For each dataset, ROC and ACC before and after data re-processing are calculated as well as the development rates.

| Base Model | Processing Method | Dataset:ABIDE | | | | | | Dataset:ABCD | | | | | |
|---|---|---|---|---|---|---|---|---|---|---|---|---|---|
| | | Before | | After | | Develop | | Before | | After | | Develop | |
| | | AUC | ACC | AUC | ACC | AUC | ACC | AUC | ACC | AUC | ACC | AUC | ACC |
| BrainNet CNN | Poly | 67.4±2.6 | 65.2±0.7 | 51.7±4.5 | 51.0±3.8 | -23.3 | -21.8 | 81.5±0.2 | 74.2±0.1 | 77.7±1.2 | 70.4±0.7 | -4.7 | -5.1 |
| | RNN | | | 52.0±3.7 | 49.0±3.5 | -22.9 | -24.9 | | | 89.0±0.5 | 80.6±0.2 | 2.9 | 9.1 |
| | TSTP | | | 58.2±3.3 | 55.4±2.5 | -13.8 | -15.0 | | | 89.7±0.5 | 81.6±0.3 | 10.0 | 10.0 |
| | Ours | | | **67.7±2.7** | **66.8±2.5** | **0.3** | **2.4** | | | **91.1±0.5** | **84.5±0.7** | **11.7** | **13.9** |
| FBNET GNN | Poly | 63.3±2.2 | 61.6±2.4 | 58.0±2.8 | 55.6±1.7 | -8.4 | -9.7 | 89.1±0.2 | 81.4±0.4 | 90.4±0.2 | 82.4±0.7 | 1.4 | 1.2 |
| | RNN | | | 56.6±3.0 | 54.2±3.1 | -10.6 | -12.0 | | | **91.1±0.6** | **83.5±1.2** | **2.8** | **2.5** |
| | TSTP | | | 64.5±3.6 | 60.2±1.7 | 1.9 | -2.3 | | | 91.0±0.5 | 82.5±0.7 | 2.1 | 1.3 |
| | Ours | | | **71.5±2.5** | **67.0±3.7** | **13.0** | **8.7** | | | 89.9±0.4 | 82.3±0.5 | 0.9 | 1.1 |
| Vanilla Transformer | Poly | 70.3±0.4 | 65.2±1.5 | 55.2±3.6 | 51.6±1.4 | -21.5 | -20.9 | 90.1±0.4 | 81.8±0.2 | 81.1±0.6 | 72.5±0.8 | -10.0 | -11.3 |
| | RNN | | | 56.6±3.4 | 59.0±3.7 | -19.5 | -9.5 | | | 86.5±0.6 | 77.5±1.1 | -4.1 | -5.1 |
| | TSTP | | | 63.5±2.3 | 58.0±2.5 | -9.7 | -11.0 | | | 82.7±2.7 | 73.0±2.6 | -8.3 | -10.8 |
| | Ours | | | **71.3±1.0** | **65.4±1.2** | **1.5** | **0.3** | | | **90.3±0.1** | **82.4±0.6** | **0.2** | **0.7** |
| BrainNet Transformer | Poly | 52.8±0.3 | 53.0±2.8 | 60.6±3.2 | 56.6±2.9 | 14.9 | 6.7 | 58.4±3.0 | 49.1±0.4 | 87.3±0.8 | 79.1±0.3 | 49.6 | 61.0 |
| | RNN | | | 50.8±2.5 | 51.6±2.8 | -3.7 | -2.6 | | | 59.5±1.2 | 53.0±1.4 | 1.9 | 7.7 |
| | TSTP | | | 58.4±1.5 | 54.0± 3.2 | 1.9 | -2.2 | | | 62.8±3.3 | 49.6±0.9 | 7.6 | 8.3 |
| | Ours | | | **72.6±3.4** | **65.4±1.0** | **37.7** | **23.4** | | | **93.6±0.3** | **85.7±0.6** | **60.3** | **74.4** |

signals. It applies GNNs on a learnable graph generated from an RNN-based signal encoder and a similarity-based graph generator [42]. (c) VanillaTransformer is a simple graph transformer based on multi-head self-attention introduced in [43]. (d) BrainNetTransformer is the SOTA model for connectome-based subject classification with an orthonormal clustering readout. [20].

*3) Metrics:* The diagnosis of ASD and prediction of biological gender are commonly evaluated tasks on ABIDE and ABCD, respectively. Because both prediction tasks are binary classification problems and both datasets are rather balanced, AUROC is a proper performance metric adopted for fair comparisons, and accuracy is applied to reflect the prediction performance when the cutoff is 0.5. In the in-depth analysis, the Root Mean Squared Error (RMSE) between reconstructed and real values is measured as we expect the reconstructed values to be as close as possible to the ground truth.. All reported performances are the average of 5 random runs on the test set with the standard deviation.

*4) Implementation Details:* We split 60% of the data for training, and 20% for validation and testing separately. In the training process of BRAINODE, Adam optimizer is used with a learning rate of $10^{-3}$. All hyperparameters are tuned using grid search. The base models are all implemented with the authors' released code with the default settings. Please refer to the Appendix for more details.

### B. Overall Performance Analysis (RQ1)

For RQ1, we hide 20% of the raw data to simulate missing values and sample 20% of the data with different frequencies to simulate sampling misalignment. We also assume the data already suffer from slightly irregular samples due to inherent instrumental errors. The raw data are re-processed by BRAINODE the and opponent models.

The results in Table I reveal that all the base models perform poorly on the raw data without any preprocessing method. This underscores the vital role of re-processing brain signals. Due to the differences in model architectures, their performances also vary. Due to the higher quality of ABCD compared to

the ABIDE, the performances of various base models are notably better on the former. After re-processing the raw data, most base models report the best performance under the re-processing of BRAINODE. On the ABIDE, the average AUC improves by 27.4%, along with a 16.9% increase in ACC. On the ABCD, the average AUC sees a boost of 15.6%, with an 8.0% increase in AUC. The base model performances with Poly are mostly the worst as it fails to capture the intricate patterns in brain signals, introducing additional noise. RNN and TSTP lead to generally better performance than Poly due to their stronger modeling of complex data, but they also add misleading information in some cases. Clearly, they lack the ability to address irregular and misaligned samples, leading to suboptimal results compared with BRAINODE. Finally, the performances of base models vary significantly, likely due to their different sensitivity to missing and misaligned data. For instance, BrainNetTransformer is relatively sensitive to missing data but more robust to noisy data, so it often performs better even against applying some of the underperforming preprocessing like Poly, where the other base models would suffer.

### C. In-Depth Analysis on Addressing the Data Challenges (RQ2)

*1) Addressing Missing Values:* To evaluate the effectiveness of BRAINODE in addressing missing values in brain signals, we compare BRAINODE with its opponents in both interpolation and extrapolation settings. The two settings correspond to two types of missing values. Interpolation means the time points of the missing value are between known data while extrapolation means the missing time points are out of the range of known data. The results with 3 and 5 missing time steps are presented in the upper half of Table II. The performance of Poly is the worst again due to its poor capability of modeling complex brain signals. BRAINODE performs best due to its strong expressiveness and capability of modeling ROI relations. Specifically, BRAINODE has the lowest RMSE, with *avg.* 74.1% drop compared with Poly, *avg.* 16.3% drop

compared with RNN, and *avg.* 74.1% drop compared with TSTP.

TABLE II: Effectiveness in addressing missing values. Results in the upper half table are analyzed in RQ2, while those in the lower half table are analyzed in RQ3. Average RMSEs are presented (the lower the better).

| Method | ABIDE | | | | ABCD | | | |
| | Interpolation | | Extrapolation | | Interpolation | | Extrapolation | |
| | 3 step | 5 step | 3 step | 5 step | 3 step | 5 step | 3 step | 5 step |
|---|---|---|---|---|---|---|---|---|
| Poly | 0.8750 | 2.025 | 0.5798 | 1.281 | 0.3573 | 1.0216 | 0.2745 | 0.6888 |
| RNN | 0.1927 | 0.2001 | 0.1940 | 0.1983 | 0.1972 | 0.1992 | 0.1525 | 0.2006 |
| TSTP | 0.1630 | 0.1648 | 0.1903 | 0.2006 | 0.1640 | 0.1599 | 0.1606 | 0.1606 |
| Ours | 0.1593 | 0.1606 | 0.1594 | 0.1600 | 0.1585 | 0.1587 | 0.1589 | 0.1592 |
| Ours-p | 0.1811 | 0.1921 | 0.1811 | 0.1970 | 0.1748 | 0.1849 | Unstable | Unstable |
| Ours-t | 0.1915 | 0.1990 | 0.1895 | 0.1958 | 0.1589 | 0.1597 | 0.1600 | 0.1601 |
| Ours-s | 0.1597 | 0.1610 | 0.1595 | 0.1601 | 0.1577 | 0.1583 | 0.1590 | 0.1594 |

*2) Addressing Irregular Samples and Sampling Misalignment:* We compare the performance of BRAINODE and its opponents on generating data across different time offsets and frequencies. Due to the lack of real continuous brain signals, we use trigonometric functions to simulate brain signals [39]. Specifically, we generate training data with regular offsets (1 point per second). Then for irregular samples, we generate testing data with irregular offsets for each sample (offsets of 0.1, 0.2 and 0.3 seconds); for sampling misalignment, we generate testing data with different frequencies (2/3, 1/2 and 1/3 seconds). The experimental results are shown in Figure 2. As offsets and frequencies grow, the performances of all methods drop. BRAINODE overperforms all other methods consistently across both settings.

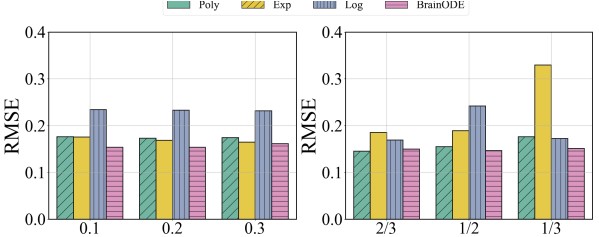

Fig. 2: Effectiveness in addressing irregular samples and sampling misalignment. Performances are grouped by offsets in the left figure and grouped by frequencies in the right figure.

### D. Ablation Study (RQ3)

To gain a deeper understanding of the contributions of the components in BRAINODE, we conduct an ablation study. We systematically evaluate the impact of removing specific components while keeping others constant. We remove the position encoder in self-attention, temporal graph, and spatial graph individually. The results are summarized in the lower half of Table 2. After removing the position encoder (denoted as ours-p), the RMSE increases *avg.* 16.9%, which shows the position encoder's importance in implying the relative time information. Without the temporal graph (denoted as ours-t), BRAINODE loses the ability to capture the temporal relation among ROIs, which causes *avg.* 10.9% increase of RMSE. As for the spatial graph, the removal (denoted as ours-s) causes *avg.* 0.1% increase of RMSE in ABIDE and some fluctuations in ABCD. The construction of a spatial graph utilized the 3D position of ROIs in the brain instead of actual structural connectivity, potentially resulting in this constrained advantage. The consideration of spatial graphs, however, makes the framework flexible enough to incorporate other types of structural connectivities such as based on structural MRI and DTI.

### E. Hyperparameter Analysis (RQ4)

We conduct a hyperparameter analysis focusing on the latent embedding size and kernel size. Our results are presented in Figure 3. As the latent embedding size increases, the RMSE rapidly decreases, indicating an improvement in the model's performance. When the latent embedding size increases further, the model's performance remains relatively stable. With the increase in kernel size, the model's RMSE decreases and then stabilizes at a certain point. The model performs well in relatively wide ranges for both hyperparameters, and too large values for both are not necessary.

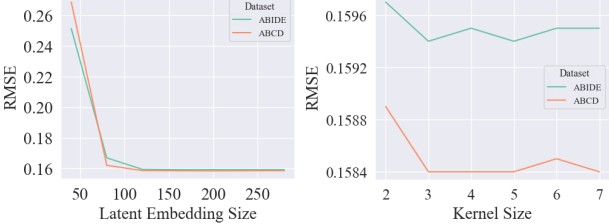

Fig. 3: Hyperparameter analysis.

### F. Real Runtime Analysis (RQ5)

To verify the practicality of our approach, we conduct experiments to record the runtime of different methods to process the same amount of data. The experimental results are shown in Table III. As Poly requires interpolation for each data segment, it has the longest processing time. The parameter number of RNN is significantly lower than that of TSTP, resulting in an average runtime of around 10.4%. The runtime of BRAINODE is at the same scale as TSTP, achieving a reasonable trade-off between effectiveness and efficiency.

TABLE III: Real runtime of different models on two datasets.

| Dataset | BrainODE | Poly | RNN | TSTP |
|---|---|---|---|---|
| ABIDE | 11.0±0.51 | 161.2±0.21 | 0.26±0.02 | 2.47±0.25 |
| ABCD | 489.87±24.22 | 3220.92±3.69 | 11.35±1.23 | 109.53±11.97 |

## V. CONCLUSION AND FUTURE WORK

In this study, we propose BRAINODE to solve the data challenges inherent in dynamic brain signals to enhance brain network analysis and downstream predictions. Leveraging a novel graph-aided neural ODE framework, our approach works by re-processing the irregular dynamic brain signals and continuously reconstructing them at any given regular time points. Through extensive experiments, we have demonstrated the efficacy of BRAINODE. The framework could be used in continuously modeling the dynamic brain signals and it could be adapted to other brain signals, such as EEG and MEG, which will be explored in future works.

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
