# OpenReview forum: "BrainODE: Dynamic Brain Signal Analysis via Graph-Aided Neural Ordinary Differential Equations"
_IEEE.org/EMBS/BHI/2024/Conference — IEEE BHI'24_

### Official Review · Reviewer_dHuG · 2024-08-07
**Reviews of Submission68**

**Overall Rating:** 7
**Confidence:** 4

**Other Quality Metrics:**

(a) Clarity of writing --- great
(b) Clinical Significance --- great
(c) Methodological Novelty --- great
(d) Experiments and Results --- great

**Questions For The Authors:**

See the weakness.

**Strengths:**

1. The use of Neural ODEs to model dynamic brain signals is a significant advancement in neuroimaging analysis.
2. Applying GCNs to incorporate structural and functional connectivity provides a deep understanding of both spatial and temporal dimensions of brain data. This dual-graph approach helps in learning robust features from the brain signals, which can enhance the detection and analysis of neural patterns relevant to clinical conditions.
3. Empirical results show better results.

**Summary Of The Paper:**

This paper presents a novel approach to addressing key challenges in the analysis of dynamic brain signals obtained from functional Magnetic Resonance Imaging (fMRI). These challenges include missing values, irregular samples, and sampling misalignment, which commonly arise due to instrumental limitations. The proposed model, BrainODE, employs Neural Ordinary Differential Equations (ODEs) to model brain signals continuously. This allows for the reconstruction of brain signals at any desired time point, thus enhancing the quality and reliability of brain network analysis used for clinical diagnostics. By integrating graph convolutional networks (GCNs) to capture spatial and temporal relationships among Regions of Interest (ROIs) within the brain, BrainODE efficiently learns the latent initial states of these ROIs, facilitating accurate and continuous signal reconstruction.

**Weaknesses:**

1. Neural ODEs need a complete data trajectory for effective dynamic modeling. Given the requirement to impute missing time series values, which can introduce bias or errors, could you clarify how you addressed this issue? Additionally, were any ablation studies conducted to explore the impact of different imputation methods on model performance?
2. Irregular sampling may compromise gradient estimation stability and accuracy due to the time integration required by neural ODEs, potentially causing numerical instabilities or inefficient learning. Could you detail any training challenges encountered during your experiments related to this issue?
3. The computational demands of processing large-scale dynamic brain signal data using Neural ODEs and GCNs might pose scalability issues, particularly with larger datasets or in real-time applications.

---

### Official Review · Reviewer_5bJE · 2024-08-10
**Review of the paper "BrainODE: Dynamic Brain Signal Analysis via Graph-Aided Neural Ordinary Differential Equations"**

**Overall Rating:** 6
**Confidence:** 3

**Other Quality Metrics:**

(a) Clarity of writing - good
(b) Clinical Significance - fair
(c) Methodological Novelty - good
(d) Experiments and Results - great

**Questions For The Authors:**

Please add more comments on the point related to the potential distance between the authors' findings and real-world clinical applications.

**Strengths:**

The proposed model enhances the usability of dynamic brain signals in clinical predictions with improved performance for various downstream brain network analysis models.
The authors report experiments on two real-world datasets to verify the effectiveness of the proposed model.

**Summary Of The Paper:**

The authors propose BrainODE, a new model designed to solve the data challenges inherent in dynamic brain signals to enhance brain network analysis and downstream predictions.

**Weaknesses:**

As stated by the authors, a limitation of their approach is the potential distance between their findings and real-world clinical applications.

---

### Official Review · Reviewer_EtME · 2024-08-17

**Overall Rating:** 7
**Confidence:** 4

**Other Quality Metrics:**

a) Clarity of writing: Great. There are some occasional typos which I mentioned earlier.

b) Clinical significance: Great. Authors express distance in between findings and real-world clinical applications.

c) Methodological Novelty: Excellent. Uses GCNs very effectively.

(d) Experiments and Results: Excellent. Uses ablation studies on top of real-world brain-imaging downstream tasks.

**Questions For The Authors:**

1. In Table 1, maybe I’m understanding the experiment incorrectly, but why are some of the AUC and ACC scores better before the re-processing step than re-processing using Poly, RNN, TSTP? For example for the BrainNet CNN and Vanilla Transformer, their performance on the ABIDE dataset before re-processing is better than 3 out of 4 re-processing techniques. It seems like reprocessing with these techniques is counterproductive for most base models.

2. Would the proposed framework be adaptable to other types of brain signals EEG and MEG?

**Strengths:**

Overall, I think the paper adequately describes the problem, the solution, and performs in-depth experimentation. Here are the specific strengths of the paper:

1. The motivation of this paper is clear in that it addresses a crucial gap with existing methods in being unable to handle the three aforementioned data challenges. Real-world data is riddled with such challenges and it is imperative to address this for downstream brain network analysis tasks.

2. The use of GCNs to model both spatial and temporal relationships among ROIs is a powerful aspect of this work.

3. End-to-end training of the framework ensures that the model components are optimized together. This can aid in model robustness.

4. The inclusion of detailed ablation studies on top of the experiments on the two real-world neuroimaging datasets adds validity to the approach. The results of the downstream tasks look very promising.

**Summary Of The Paper:**

This paper introduces a novel approach to brain network analysis by addressing 3 key challenges associated with dynamic brain signals (missing values, irregular samples, and sampling misalignment). The proposed BrainODE method leverages Neural Ordinary Differential Equations (ODEs) to continuously model these signals by learning latent initial values and ODE functions, enabling effect signal reconstruction at any time point. The framework includes components for brain latent state learning using CNNS and self-attention mechanisms, and temporal and spatial graph construction to capture relationships among brain regions of interest (ROI), with Graph Convolutional Neural Networks (GCN). The model is trained end-to-end in an autoencoder framework.  Experimental results on two datasets (ABIDE and ABCD) demonstrate BrainODE’s superior performance in classification and handling data irregularities.

**Weaknesses:**

Here are some weaknesses of the paper:

1. This work does not mention anything about the variability in brain structure across different subjects. This variability can affect the generalizability of models trained on specific datasets.

2. From the ablation studies, the removal of the spatial graph only causes a 0.1% increase of RMSE in the ABIDE and some minor fluctuations in ABCD. This seems fairly insignificant compared to the other components in the pipeline. I think this aspect needs to be further explored as it might add unnecessary complexity to the framework without much-added benefit.

3. This weakness is probably overstated in such studies, but it’s worth mentioning anyway. The paper does not address how the complex outputs of brain ODE, particularly the latent representations and ODE-based reconstructions, can be interpreted in a way that is meaningful and actionable for clinicians. Moreover, how can you validate these latent representations?

List of typos:

1. "Exsiting" (page 2, section II, Backgrounds and Related Works) - should be corrected to "Existing."

2."mode learning" (page 2, section II, Backgrounds and Related Works) to "model learning."

3."Misalignmnet" (page 3, Figure 1 caption)  to "Misalignment."

4."Acitvate Function" (page 3, Figure 1 caption) to "Activate Function."

5. "additonal" (page 4, section D, Brain Temporal Relation and Spatial Relation Incorporation) to "additional."

6. "tempora" (page 4, section D, Brain Temporal Relation and Spatial Relation Incorporation, paragraph 1) to "temporal."

7."Evoluation" (page 2, section II, Backgrounds and Related Works) to "Evolution."

8."Acitvate Function" (page 4, Figure 1) to "Activate Function."

---

### Decision · Program_Chairs · 2024-09-23

Accept